# MTGS: A Novel Framework for Multi-Person Temporal Gaze Following and Social Gaze Prediction

**Anshul Gupta   Samy Tafasca   Arya Farkhondeh   Pierre Vuillecard   Jean-Marc Odobez**

Idiap Research Institute, Martigny, Switzerland

École Polytechnique Fédérale de Lausanne, Switzerland

`{agupta, stafasca, afarkhondeh, pvuillecard, odobez}@idiap.ch`

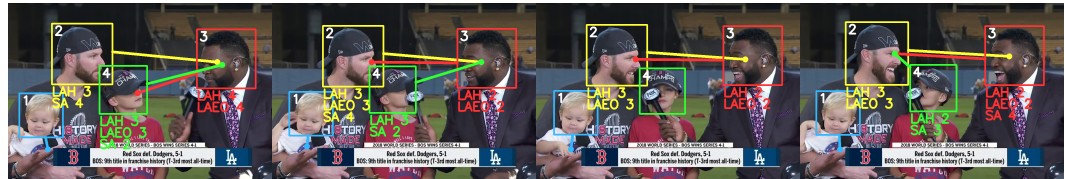

Figure 1: Results of our multi-person and temporal transformer architecture for joint gaze following and social gaze prediction, namely Looking at Humans (LAH), Looking at Each Other (LAEO), and Shared Attention (SA). For each person, the social gaze predictions are listed with the associated person ID (*e.g.* in frame 1, person 2 is in SA with person 4). More qualitative results can be found in the supplementary G.

## Abstract

Gaze following and social gaze prediction are fundamental tasks providing insights into human communication behaviors, intent, and social interactions. Most previous approaches addressed these tasks separately, either by designing highly specialized social gaze models that do not generalize to other social gaze tasks or by considering social gaze inference as an ad-hoc post-processing of the gaze following task. Furthermore, the vast majority of gaze following approaches have proposed models that can handle only one person at a time and are static, therefore failing to take advantage of social interactions and temporal dynamics. In this paper, we address these limitations and introduce a novel framework to jointly predict the gaze target and social gaze label for all people in the scene. It comprises (i) a temporal, transformer-based architecture that, in addition to frame tokens, handles person-specific tokens capturing the gaze information related to each individual; (ii) a new dataset, VSGaze, built from multiple gaze following and social gaze datasets by extending and validating head detections and tracks, and unifying annotation types. We demonstrate that our model can address and benefit from training on all tasks jointly, achieving state-of-the-art results for multi-person gaze following and social gaze prediction. Our annotations and code will be made publicly available.

## 1   Introduction

Social interaction plays a pivotal role in our daily lives and is influenced by an array of behavioral elements, encompassing not only verbal communication but also non-verbal cues like gestures (53) or body language. In particular, the ability to decode people's gaze, including communicative behaviors like eye contact and shared attention on a particular object, is highly related to our capacity to connect with or learn from others (44). In contrast, the absence or impairment of these skills is often indicative of developmental disorders such as autism (42). Thus, designing social gaze prediction algorithms

38th Conference on Neural Information Processing Systems (NeurIPS 2024).

and systems has attracted considerable attention from different communities, ranging from medical diagnosis to human-robot interactions (42; 43; 36).

In this work, we investigate whether we can build a unified framework to infer from video data the *gaze target* and *social gaze label* in *one stage* for *all people* in the scene. This requires: (i) A new architecture capable of jointly modeling these tasks, (ii) A large-scale dataset with annotations for all the tasks. Specifically, we focus on the LAH, LAEO and SA social gaze tasks (illustrated in Fig. 1).

Methods for social gaze prediction in the literature adopt one of two approaches. The first one focuses on the design of dedicated networks to process pairs of head crops and potentially other scene information (31; 30; 11; 5; 12; 46). While their specialization makes them effective, they offer little room for generalization to other gaze-related tasks. The second one first address the gaze following tasks (40), defined as predicting the 2D location people's gaze targets, and then uses the predicted gaze points to infer social gaze through ad-hoc post-processing schemes. For instance, combining gaze following heatmaps from multiple people to predict shared attention (9).

However, gaze following itself is a challenging task. Besides geometric aspects, the task requires understanding and establishing a correspondence between top-down information related to the person's state, activity, cognitive intent, and bottom-up saliency related to the scene context (salient items like objects or talking people). Furthermore, gaze following performance, as measured by distance, does not always translate to similar social semantic performance (*e.g.* when evaluating if the predicted gaze point falls on a person's head or not (48)).

**Motivation.** Existing methods for gaze following suffer from several drawbacks. First, most of the methods perform prediction for a single person (40; 41; 9; 14; 17; 48), requiring multiple inference passes on the same image to process multiple people in the scene. In contrast, a multi-person gaze following architecture processes the image only once and has to capture salient items for all people in the scene, while maintaining the ability to infer the gaze target of each individual. This is inherently more complex and challenging. Another drawback of the single-person formulation is that it does not explicitly model people's interactions, thereby preventing the possibility of jointly inferring people's gaze target and social gaze attributes. Secondly, the majority of proposed models for gaze following are static, using only a single image at a time. This is partly due to the absence of large and diverse video datasets, and the difficulty of leveraging large-scale static ones like GazeFollow (41). This is a limitation, as temporal information can capture head and gaze coordination patterns (43) which can help gaze direction inference, especially when the eyes are not completely visible (34).

Finally, none of these methods have investigated learning the gaze following and social gaze prediction tasks jointly. Thus, it remains a research question whether such formulation can improve performance by having social cues inform the gaze following task and vice-versa, or if performance would degrade as we try to accommodate multiple tasks, datasets, and people within the same framework.

**Contributions.** Given these motivations, we propose a new, unified framework for gaze following and social gaze prediction with the following contributions:

- A novel temporal and multi-person architecture for gaze following and social gaze prediction (Sec. 3). Our approach posits people as specific tokens that can interact with each other and the scene content (*i.e.* frame tokens). This token-based multi-person representation allows for the modeling of (i) temporal information at multiple levels (from 2D gaze direction to 2D gaze target level), (ii) the joint prediction of the gaze target and social gaze label.
- A new dataset, VSGaze, that unifies annotation types across multiple gaze following and social gaze datasets (Sec. 4.1.1).
- New social gaze protocols and metrics for better evaluating semantic gaze following performance (Sec. 4.3).

**Results.** In our experiments (Sec. 5), we show that our architecture achieves state-of-the-art results for multi-person gaze following, while also performing competitively against single-person models. It is also able to leverage our proposed VSGaze dataset to jointly tackle gaze following and social gaze prediction, achieving competitive performance compared to methods trained on individual tasks. In particular, our experiments further demonstrate that the performance *benefits from this joint prediction,* i.e. *adding the social loss, improves gaze following performance, and vice-versa.* Finally, the new social gaze metrics provide complementary information to the standard distance-based metrics, helping assessing model performance from the social interaction perspective.

It is worth noting that our architecture is easily extendable and allows for the integration of auxiliary person-specific information that can influence the final predictions. In the supplementary F, we explore this aspect by integrating people's speaking status in the person tokens to improve the results.

## 2   Related work

**Gaze Following.** Typical methods for this task exploit a two-branch architecture: one for processing the scene and the other for processing the person of interest (40; 9; 14; 17; 22; 23; 26). They have distinguished themselves by the addition of other relevant modalities like depth (14; 17; 23), pose (17), and objects (20), or by potentially leveraging scene geometry (19; 23; 14). However, only a few efforts have addressed the multi-person case. (22) first proposed a simple architecture relying on a scene backbone to get a person-agnostic image representation that is subsequently fused with the head crop features of each individual obtained using another backbone. While this reduces computation, the model does not account for person interactions, as each head is processed separately. In another direction, (52; 51) rely on a transformer-based architecture to perform multi-person gaze following. Their methods borrow from DETR (6), taking the image as input and simultaneously predicting the head bounding box and gaze target for every person in the scene. While these methods can implicitly model person interactions, an important limitation is that they compute performance on detected heads which are matched to the ground truth. Given that both the head detection and matching steps are error-prone, it precludes comparing their results to others. We provide examples in the supplementary (Fig. 8) where (51) makes incorrect head detections.

**Temporal gaze estimation.** Temporal information has proven effective for 3D gaze estimation. Previous research developed models to learn from various inputs, including face, eyes, and facial landmarks using a multi-stream recurrent CNN (37); eyes and visual stimuli or raw RGB frames from in-the-wild settings using convolutional RNNs or LSTMs (38; 25); and the temporal coordination of gaze, head, and body orientations using LSTMs (34). However, the use of such methods for the gaze following task in arbitrary scenes has been underexplored. The only exceptions are (9) who introduced a convolutional LSTM block at the bottleneck of the heatmap prediction architecture, and (32) who leveraged temporal attention over aggregated frame-level features. However, both approaches only showed a slight improvement compared to their static versions, highlighting the challenge of exploiting temporal information for this task. Conceptually, the methods did not model 2D gaze direction dynamics, and can not be extended for multi-person gaze inference.

**Social gaze prediction.** Several research papers in the literature are dedicated to the study of looking at each other (LAEO) and shared attention (SA) tasks. For LAEO, most methods rely on processing the head crops to obtain some gaze directional information, and then combining it with 2D or inferred 3D geometric information to predict the LAEO label (31; 11; 30; 5). Drawbacks include processing pairs of persons independently and, as they only process heads and do not address gaze following, lacking global image context and not extending easily to other social tasks like SA. Similarly to (52; 51), a recent paper (15) proposed an encoder-decoder transformer architecture to predict heads and the LAEO labels, and while achieving good results, suffers from the same drawbacks.

Regarding shared attention, the first method to address it in the wild was (12), which framed the problem as 2 tasks: the binary classification of whether SA occurs in a frame, and the inference of the location of the SA target object. Their method combined predicted 2D gaze cones of people in the scene with a heatmap of object region proposals, while others (46) directly inferred SA from the raw image. Since then, several methods leveraged combining gaze following heatmap predictions of all people (9; 52) and improved performance. Nevertheless, the above task formulation (12) used by all papers suffers from two main issues: (i) it cannot distinguish between multiple SA instances if they occur in the same frame; (ii) it does not determine which specific people are sharing attention. Our work solves both problems by framing the task as a binary classification between pairs of people. This formulation is more natural and has the benefit of extending to other social gaze tasks.

Finally, none of the previous works performed both social gaze prediction tasks. There are three notable and interesting exceptions. First, (13) who addressed the inference of gaze communication activities (atomic and events, including LAEO and SA) using a graph-based approach with 12 dimensional tokens. However, as it predicts a single gaze 'state' for each person, it is problematic as it does not allow for simultaneous LAEO and SA. It also does not identify the other person involved in the social gaze interaction. The second is (7) who addressed dyadic communication by proposing a gaze following style 2-branch architecture processing order-dependent pairs of people. However, inference using (7) is inefficient because the model needs $\frac{N_p!}{(N_p-2)!}$ forward passes to consider all pairwise relationships for a given scene with $N_p$ people. Also both methods do not address the gaze following task. More recently, (16) did address all tasks using a graph approach, but there was no temporal modeling nor joint training on all tasks.

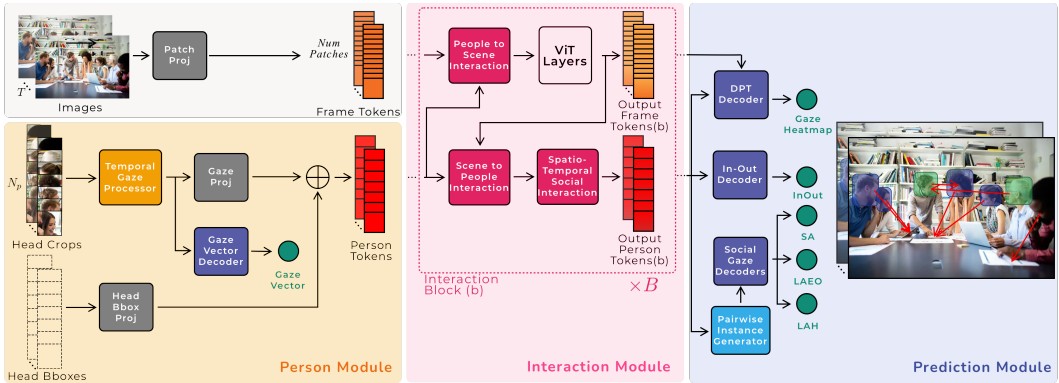

Figure 2: Proposed architecture for multi-person temporal gaze following and social gaze prediction. See approach overview in Section 3.

## 3 Architecture

As motivated in Section 1, designing a temporal architecture for gaze following is challenging given the lack of large scale video datasets. Since the broader scene tends to remain relatively static for a short temporal window, we focused on modeling person-level temporal information to simplify the learning problem. In particular, our architecture processes person and scene tokens through separate transformers, with a temporal transformer for processing the person tokens. At the same time, it facilitates interactions between the scene and person tokens via cross-attention.

**Approach overview.** Our approach is illustrated in Fig. 2. It takes as input a sequence of $t = 1 \ldots T$ frames, as well as the head bounding box tracks $\mathbf{h}^{\text{box}}_{i,1:T}$ and corresponding head crops $\mathbf{h}^{\text{crop}}_{i,1:T}$ which are assumed to have been extracted for each of the $i = 1 \ldots N_p$ persons. The outputs are the sequence of gaze heatmaps $\mathcal{A}_{i,1:T}$ and in-out gaze labels $\mathbf{o}_{i,1:T}$ for each person $i$, as well as the sequence of per-frame pair-wise social gaze labels for each task and pair $i, j \in \{1 \ldots N_p\}$: $\mathbf{e}_{i \rightarrow j,1:T}$ for LAH, $\mathbf{e}_{i \leftrightarrow j,1:T}$ for LAEO, and $\mathbf{c}_{i,j,1:T}$ for SA.

The model proceeds as follows. First, each frame $t$ is processed by a standard ViT tokenizer to produce the set of patch-wise frame tokens $\mathbf{f}_t$, resulting in a sequence of frame tokens $\mathbf{f}_{1:T}$. In parallel, the Person Module processes the sequence of head crops from each person $i$ using the Temporal Gaze Processor, and the resulting sequence outputs are then tokenized at each frame along with the bounding box locations to produce the sequence of person token $\mathbf{p}_{i,1:T}$. Secondly, the Interaction Module jointly processes the frame and person tokens, iteratively updating them at each time step through person-scene cross-attention interaction components and scene ViT self-attention, and in time through person spatio-temporal social interaction components. Finally, the Prediction Module processes at each time step the resulting frame and person tokens (from multiple blocks) to infer the sequence of gaze heatmaps and in-out gaze labels for each person, as well as pair-wise social gaze labels. We detail the three modules in the next sections.

### 3.1 Person Module

This module aims to model person-specific information relating to gaze and head location.

**Temporal Gaze Processor.** It aims to capture all gaze-related information (direction, dynamics). First, individual head crops $\mathbf{h}^{\text{crop}}_{i,t}$ are processed by a Gaze Backbone $\mathcal{G}_{\text{stat}}$ to produce gaze embeddings according to $\mathbf{g}^{\text{stat}}_{i,t} = \mathcal{G}_{\text{stat}}(\mathbf{h}^{\text{crop}}_{i,t})$. Then, to model the gaze dynamics of a person, we rely on a Temporal Gaze Encoder $\mathcal{G}_{\text{temp}}$ to process the sequence $\mathbf{g}^{\text{stat}}_{i,1:T}$ of gaze embeddings plus learnable temporal position embeddings $\mathbf{x}_{1:T}$ and obtain their temporal counterparts: $\mathbf{g}^{\text{temp}}_{i,1:T} = \mathcal{G}_{\text{temp}}(\mathbf{g}^{\text{stat}}_{i,1:T} + \mathbf{x}_{1:T})$. $\mathcal{G}_{\text{temp}}$ is implemented as a single Transformer layer with self-attention. Finally, to supervise the learning of relevant gaze embeddings, we attach a Gaze Vector Decoder that predicts a person's 2D gaze vector at each time step, $\mathbf{g}^{\text{v}}_{i,t} = \mathcal{G}_{\text{vec}}(\mathbf{g}^{\text{temp}}_{i,t})$, where $\mathcal{G}_{\text{vec}}$ is implemented as a 2-layer MLP.

**Person tokenization.** The person tokens are obtained by projecting the temporal gaze embeddings and normalized 4d head box locations using learnable linear layers ($\mathcal{P}_{\text{gaze}}$ and $\mathcal{P}_{\text{box}}$ respectively) to tokens of same dimension than frame token, and adding them together:

$$\mathbf{p}_{i,t} = \mathcal{P}_{\text{gaze}}(\mathbf{g}_{i,t}^{\text{temp}}) + \mathcal{P}_{\text{box}}(\mathbf{h}_{i,t}^{\text{box}}). \tag{1}$$

## 3.2 Interaction Module

The Interaction module aims at modeling the exchange of information between persons and the scene at each time step, as well as the spatio-temporal social interactions between people. One important goal of this process is to align the person and frame token representations so that (i) *person-specific* gaze heatmaps can be predicted from the set of output frame tokens and each person output token; (ii) in-out gaze and social gaze prediction can be made from the person tokens.

To do so, we designed the module to consist of $B$ blocks, each comprising Person-Scene Interaction and Spatio-Temporal Social Interaction components. The input to the first block is the set of person tokens $\mathbf{p}_{1:N_p,1:T}$ from the Person Module, and the frame tokens $\mathbf{f}_{1:T}$. Each block then processes the set of output[1] person tokens $\mathbf{p}_{1:N_p,1:T}^{\text{o,b-1}}$ and output frame tokens $\mathbf{f}_{1:T}^{\text{o,b-1}}$ from the previous block, and returns updated tokens after a series of self/cross-attention layers through the components.

**Person-Scene Interaction.** This component models the interactions between people and the scene and can capture inferring gaze to scene objects or body parts like hands or exploit some global context. It is inspired by ViT-Adaptor (8) which has shown good performance for dense prediction tasks when relying on pretrained models and small amounts of data for the target task. It proceeds in 3 steps:

(i) People-to-Scene Encoder $\mathcal{I}_{\text{ps}}^{\text{b}}$: it updates the frame tokens with person information relevant to gaze by processing the frame tokens $\mathbf{f}_t^{\text{o,b-1}}$ and frame-level person tokens $\mathbf{p}_{1:N_p,t}^{\text{o,b-1}}$ according to $\mathbf{f}_t^{\text{p,b}} = \mathcal{I}_{\text{ps}}^{\text{b}}(\mathbf{f}_t^{\text{o,b-1}}, \mathbf{p}_{1:N_p,t}^{\text{o,b-1}})$. It is implemented as a single Transformer layer with cross-attention, where $\mathbf{f}_t^{\text{o,b-1}}$ generate the queries and $\mathbf{p}_{1:N_p,t}^{\text{o,b-1}}$ generate the keys and values.

(ii) The updated frame tokens $\mathbf{f}_t^{\text{p,b}}$ pass through the standard set of of ViT layers $\mathcal{V}_b$ to process the scene information, resulting in the output frame tokens for the block $b$: $\mathbf{f}_t^{\text{o,b}} = \mathcal{V}_b(\mathbf{f}_t^{\text{p,b}})$.

(iii) Scene-to-People Encoder $\mathcal{I}_{\text{sp}}^{\text{b}}$: it updates the person tokens so that they capture location information related to the salient items they are probably looking at. It works by processing the frame-level person tokens $\mathbf{p}_{1:N_p,t}^{\text{o,b-1}}$ and obtained frame tokens $\mathbf{f}_t^{\text{o,b}}$ according to: $\mathbf{p}_{1:N_p,t}^{\text{s,b}} = \mathcal{I}_{\text{sp}}^{\text{b}}(\mathbf{p}_{1:N_p,t}^{\text{o,b-1}}, \mathbf{f}_t^{\text{o,b}})$. It is also implemented as a single Transformer layer with cross-attention, where the set $\mathbf{p}_{1:N_p,t}^{\text{o,b-1}}$ generates the queries and $\mathbf{f}_t^{\text{o,b}}$ generates the keys and values.

**Spatio-temporal Social Interaction.** This component allows the sharing of information between people and of the alignment of their representations for social gaze prediction. This also include modeling the temporal evolution of individual tokens. To achieve this, a Social Encoder $\mathcal{I}_{\text{pp}}^{\text{b}}$ first processes and updates the frame-level person tokens $\mathbf{p}_{1:N_p,t}^{\text{s,b}}$ to capture interactions between people at each frame, according to: $\mathbf{p}_{1:N_p,t}^{\text{p,b}} = \mathcal{I}_{\text{pp}}^{\text{b}}(\mathbf{p}_{1:N_p,t}^{\text{s,b}})$. It is followed by a Temporal Person Encoder $\mathcal{I}_{\text{pt}}^{\text{b}}$ that processes the updated person token sequences $\mathbf{p}_{i,1:T}^{\text{p,b}}$ of each person $i$ and updates them to capture temporal patterns of attention, resulting in the output person tokens for the block: $\mathbf{p}_{i,1:T}^{\text{o,b}} = \mathcal{I}_{\text{pt}}^{\text{b}}(\mathbf{p}_{i,1:T}^{\text{p,b}})$. Both $\mathcal{I}_{\text{pp}}^{\text{b}}$ and $\mathcal{I}_{\text{pt}}^{\text{b}}$ are implemented as a single Transformer layer with self-attention.

## 3.3 Prediction Module

The Prediction Module processes the set of output person $\{\mathbf{p}_{1:N_p,1:T}^{\text{o,b}}\}$ and frame $\{\mathbf{f}_{1:T}^{\text{o,b}}\}$ tokens from all Interaction Module blocks to predict the person-specific gaze heatmaps and in-out labels, as well as the pair-wise social gaze labels.

**Gaze Heatmap Prediction.** Here, we follow the model introduced in (49) which takes inspiration from the DPT decoder (39) for dense prediction tasks, and adapts it to handle multiple heatmap predictions from the same ViT outputs. This is performed *by conditioning the decoding on each person's token*. The standard DPT decodes the image features from multiple layers of a ViT in a Feature Pyramid Network (27) style. It works by fusing at block level $b$ the feature maps from level $b + 1$ after an upsampling stage, and the feature maps computed by a reassemble stage from the ViT output of block $b$. We aim to apply this approach to the frame tokens $\{\mathbf{f}_{1:T}^{\text{o,b}}, b = 1 : B\}$,

---

[1]Note that the superscript $o$, $p$ and $s$ do not represent indices, but intermediate token updates within a block $b$.

but conditioned on a specific person. In our model, this is achieved through a modification in the reassemble stage, in which the image feature maps produced by the standard reassemble stage are multiplied at every location (using a Hadamard product) with the projected person token $\mathbf{p}_{i,t}^{o,b}$ of that same block level. The gaze heatmap $\mathcal{A}_{i,t}$ for each person at each frame is thus obtained as:

$$\mathcal{A}_{i,t} = \mathcal{D}(\{(\mathbf{f}_t^{o,b}, \mathbf{p}_{i,t}^{o,b}), b = 1 : B\}) \tag{2}$$

where $\mathcal{D}$ denotes this conditional DPT. See (49) and supplementary H for details.

**Social Gaze Prediction.** This decoder processes the person tokens from all $B$ Interaction Module blocks to predict the social gaze label for every pair of people in every frame. In practice, the $B$ tokens $\{\mathbf{p}_{i,t}^{o,1} \ldots \mathbf{p}_{i,t}^{o,B}\}$ corresponding to a single person in a frame are linearly projected and concatenated to produce a multi-scale person token $\mathbf{p}_{i,t}^{ms}$. Then, to predict a social gaze label, pairs of these tokens are concatenated and processed by the decoders $E$ for LAH and $C$ for SA (illustrated through the Pairwise Instance Generator in Fig. 2). Their outputs are the predicted LAH score $\mathbf{e}_{i \to j,t}$ for person $i$ looking at $j$, and the predicted SA score $\mathbf{c}_{i,j,t}$ for $i, j$.

$$\mathbf{e}_{i \to j,t} = E(\mathbf{p}_{i,t}^{ms}, \mathbf{p}_{j,t}^{ms}) \text{ and } \mathbf{c}_{i,j,t} = C(\mathbf{p}_{i,t}^{ms}, \mathbf{p}_{j,t}^{ms}). \tag{3}$$

$E$ and $C$ are implemented as 3-layer MLPs with residual connections. For LAEO, both people $i, j$ need to be looking at each other for a positive label, and either one can be looking away for a negative label. Hence, we simply compute the LAEO score $\mathbf{e}_{i \leftrightarrow j,t}$ as $\min(\mathbf{e}_{i \to j,t}, \mathbf{e}_{j \to i,t})$.

**In-Out Prediction.** This decoder $\mathcal{O}$ processes the multi-scale person tokens $\mathbf{p}_{i,t}^{ms}$ to predict at every frame whether people are looking inside the frame or outside the frame, as $\mathbf{o}_{i,t} = \mathcal{O}(\mathbf{p}_{i,t}^{ms})$, where $\mathcal{O}$ is implemented as a 5-layer MLP with residual connections.

### 3.4 Losses

The total loss $\mathcal{L}$ is a linear combination of the gaze heatmap loss $\mathcal{L}_{HM}$, gaze vector loss $\mathcal{L}_{VEC}$, social gaze losses $\mathcal{L}_{LAH}, \mathcal{L}_{SA}$ and the in-out loss $\mathcal{L}_{IO}$:

$$\mathcal{L} = \lambda_{HM}\mathcal{L}_{HM} + \lambda_{VEC}\mathcal{L}_{VEC} + \lambda_{LAH}\mathcal{L}_{LAH} + \lambda_{SA}\mathcal{L}_{SA} + \lambda_{IO}\mathcal{L}_{IO} \tag{4}$$

$\mathcal{L}$ is applied at each time step per person for $\mathcal{L}_{HM}, \mathcal{L}_{VEC}, \mathcal{L}_{IO}$, and per pair for $\mathcal{L}_{LAH}, \mathcal{L}_{SA}$. All losses are standard: $\mathcal{L}_{HM}$ is defined as the pixel-wise MSE loss between the GT and predicted heatmaps, $\mathcal{L}_{VEC}$ as the cosine loss, and the social gaze and in-out losses as binary cross-entropy losses. Since LAEO is inferred from LAH predictions (Sec 3.3), we do not have any LAEO loss.

## 4 Experiments

### 4.1 Datasets

We perform experiments on multiple gaze following and social gaze datasets.

**GazeFollow (41)** is a large-scale static dataset for gaze following, featuring 122K images. Most images are annotated for a single person with their head bounding box and gaze target point. The test set contains gaze point annotations by multiple annotators. Despite lower quality images and annotations, given its rich diversity, it remains a good dataset to use for pre-training.

**VideoAttentionTarget (VAT) (9)** is a video dataset, annotated with head bounding boxes, gaze points, and inside vs outside frame gaze for a subset of the people in the scene. It contains 1331 clips collected from 50 shows on YouTube.

**ChildPlay (48)** is a recent video dataset for gaze following, annotated with head bounding boxes, gaze points, and a label indicating 7 non-overlapping gaze classes including inside frame, outside frame and gaze shifts. It contains 401 clips from 95 YouTube videos, and features children playing and interacting with other children and adults.

**VideoCoAtt (12)** is a video dataset for shared attention estimation, containing 380 videos or 492k frames from TV shows. When a shared attention behavior occurs (i.e. about 140k frames), the relevant frames are annotated with the bounding box of the target object, as well as the head bounding boxes of the people involved.

**UCO-LAEO (31)** is a video dataset for LAEO estimation, annotated with head bounding boxes, and a label indicating whether two heads are LAEO. It contains 22,398 frames from 4 TV shows.

| Dataset | Gaze Points | LAH | LAEO | SA |
|---------|-------------|-----|------|-----|
| GazeFollow (41) | 118k | 27k/493k | 0 | 0 |
| VAT (9) | 109k | 74k/729k | 13k/461k | 16k/94k |
| ChildPlay (48) | 217k | 59k/682k | 7k/351k | 4k/55k |
| VideoCoAtt (12) | 367k | 290k/1551k | 0 | 400k/918k |
| UCO-LAEO (31) | 21k | 21k/36k | 10k/54k | 0 |
| **VSGaze** | 714k | 444k/2998k | 30k/866k | 420k/1067k |

Table 1: Person-wise gaze point and pair-wise social gaze annotation (positive/negative) statistics for our datasets. VSGaze unifies annotation types across VAT, ChildPlay, VideoCoatt and UCO-LAEO.

Two other interesting datasets with annotations for multiple social gaze behaviours are VACATION (13) and GP-static (7). However, VACATION does not provide annotations for all social gaze behaviours when they occur simultaneously (examples in supplementary Fig. 7). This annotation scheme is problematic and is linked to their method design as described in Sec. 2. On the other hand, GP-static only considers dyadic interactions and is not publicly available.

### 4.1.1 VSGaze dataset

A limitation of the above datasets is that they only contain annotations for gaze following or specific social gaze tasks. Hence, we propose the **V**ideo dataset with **S**ocial gaze and **Gaze** following annotations or **VSGaze** dataset. VSGaze extends head track annotations and unifies annotation types across VAT, ChildPLay, VideoCoAtt and UCO-LAEO. This allows for joint training of gaze following and social gaze, and provides new tasks and metrics for evaluating performance on the component datasets. The construction of VSGaze is described below.

**Extending Head Track Annotations.** As each dataset contains annotations for only a subset of people in the scene, we detect all missing heads using the pre-trained Yolov5 head detection model (24) used by Tafasca *et al*. (48), and track them using the ByteTrack algorithm (54). We further manually verified the accuracy of the obtained tracks. This step is vital to obtain positive and negative social gaze pairs. Consider a scene with 3 people, $i, j, k$, where $i$ is looking at $j$. If only $i$ is annotated, the positive LAH pair $i \rightarrow j$ would be missed. The negative LAH pair $i \rightarrow k$ would also be missed.

**Unifying Gaze Following and Social Gaze Annotations.** Given the extended set of head bounding box annotations, as well as existing gaze following and social gaze annotations, we then process these annotations to obtain gaze following and social gaze labels across all the datasets. The gaze target for UCO-LAEO and VideoCoAtt is set as the center of the LAEO and SA target bounding box respectively. LAH pairs are obtained by checking if the gaze target falls inside another person's head box. For LAEO we check the reverse as well. SA pairs are obtained by checking if the gaze targets for both people fall in the same head box. We provide the detailed protocol in the supplementary B.

**Annotation statistics** are summarized in Table 1. Overall, VideoCoatt is the largest source of annotations except for LAEO. We also see that the pair-wise annotations are skewed towards negative cases. The statistics further provide insight into the content of the datasets. As VAT, VideoCoAtt and UCO-LAEO contain clips from TV shows, there are many more instances of looking at other people and at each other. On the other hand for ChildPlay, LAH mainly occurs when the supervising adult looks at a child and there is limited LAEO.

### 4.2 Training and Validation

We follow standard practice (9; 14; 17) by first training the static version of our model (i.e. with no temporal attention $\mathcal{G}_{\text{temp}}$, $\mathcal{I}_{\text{pt}}^{\text{b}}$) on GazeFollow. It is trained for 20 epochs with a learning of rate of $1 \times 10^{-4}$. The resulting weights then serve as initialization for our proposed temporal model. We freeze the ViT $\mathcal{V}$ and train the temporal model on VSGaze for another 20 epochs with a learning rate of $3 \times 10^{-6}$. For validation, we use the provided splits for UCO-LAEO, VideoCoAtt and ChildPlay, and the splits proposed by Tafasca *et al*. (48) for GazeFollow and VAT. For all our experiments, we use $T = 5$ frames with a temporal stride of 3. To allow for batch training, we randomly sample up to $N_p = 4$ people in a scene (padding in case there are less). During testing, we evaluate per sample and consider all people. The Interaction Module consists of $B = 4$ blocks, interacting with $\mathcal{V}$ at layers $\{2, 5, 8, 11\}$. Additional implementation details are provided in the supplementary E.

| Method | PP | Dist. $\downarrow$ | $AP_{IO}$ $\uparrow$ | $F1_{LAH}$ $\uparrow$ | $F1_{LAEO}$ $\uparrow$ | $AP_{SA}$ $\uparrow$ |
|---|---|---|---|---|---|---|
| $\text{Chong}_S$* (9) | ✓ | 0.121 | 0.918 | 0.778 | 0.562 | 0.288 |
| $\text{Chong}_T$* (9) | ✓ | 0.130 | **0.956** | 0.764 | 0.529 | 0.331 |
| Gupta* (17) | ✓ | 0.119 | 0.929 | 0.784 | 0.590 | 0.335 |
| Ours-noGF | ✗ | - | - | 0.738 | 0.579 | 0.515 |
| Ours-noSoc | ✓ | 0.111 | 0.945 | 0.802 | 0.598 | 0.339 |
| Ours | ✗ | **0.107** | 0.940 | 0.795 | 0.590 | **0.576** |
| Ours-PP | ✓ | **0.107** | 0.940 | **0.812** | **0.603** | 0.352 |

Table 2: Comparison against gaze following methods on VSGaze. All models were trained on VSGaze. PP indicates social gaze predictions from post-processing gaze following outputs (✓) vs predictions from decoders (✗). Best results are in bold, second best results are underlined.

## 4.3 Evaluation

**Gaze Following.** We use the standard metrics:

- *AUC*: for GazeFollow, the predicted heatmap is compared against a binary GT map with value 1 at annotated gaze point positions, to compute the area under the ROC curve.
- *Distance (Dist.)*: the arg max of the heatmap provides the gaze point. We can then compute the L2 distance between the predicted and GT gaze point on a $1 \times 1$ square. For GazeFollow, we compute Minimum (Min.) and Average (Avg.) distance against all annotations.
- *In-Out AP ($AP_{IO}$)*: it is the Average Precision (AP) of the In-Out gaze prediction scores.

**Social Gaze.** We propose an evaluation protocol for LAH, as well as a new pair-wise evaluation protocol for SA as motivated in Sec. 2.

*Social gaze decoders:* For the LAH task we compute an F1 score. A sample is an individual person $i$, and at inference, it is assigned the person $\hat{j}$ for which the $\mathbf{e}_{i \to j,t}$ is the largest. Hence, for a GT positive case, the prediction will be considered as a true positive if $\hat{j}$ matches the GT target AND $\mathbf{e}_{i \to \hat{j},t}$ is above 0.5. Otherwise, the prediction is a false negative. Similarly, a GT negative is a true negative if $\mathbf{e}_{i \to \hat{j},t}$ is below 0.5, otherwise it is a false positive. For LAEO, as with LAH, we compute an F1 score. A sample is a pair of people $i, j$, and for person $i$, we consider $\mathbf{e}_{i \leftrightarrow \hat{j},t}$ with the highest score. We then set $\mathbf{e}_{i \leftrightarrow j,t}$ to 0 $\forall j \neq \hat{j}$ before computing the performance. We also do the reverse for $j, i$. For SA, we compute a standard AP score by considering a sample as a pair of people, and thresholding predicted scores at different values to compute the area under the Precision-Recall curve.

*Gaze following methods:* Social gaze predictions are obtained by post-processing the predicted gaze points. For LAH, we check whether the predicted gaze point for a person falls inside the target person's head box. For LAEO we check the reverse as well. We compute F1 scores for both. For SA, we check if the distance between two people's predicted gaze points is within a set of thresholds and compute an AP score.

## 5 Results

### 5.1 Comparison against the State-of-The Art

We compare against recent SoTA methods addressing either social gaze tasks or gaze following. In addition, for fairness and to evaluate the impact of the VSGaze dataset, we also re-trained on this dataset the static image based models of Chong (9) ($\text{Chong}_S$*) and Gupta (17) (Gupta*), as well as the temporal model of (9) ($\text{Chong}_T$*), the only temporal gaze following model with available code.

**VSGaze.** The results on VSGaze are given in Table 2. Note that regarding our approach, for social gaze, we compute the scores by leveraging either the predictions from the respective task decoders (Ours), or by post-processing the gaze following outputs of our model (Ours-PP).

Compared to the baselines, we observe that our model achieves the best performance for all tasks except for in-out gaze prediction. In particular, we achieve significant gains in the distance and $AP_{SA}$ metrics when leveraging the predictions from the SA decoder. The latter highlights the importance of modeling SA as a classification task compared to post-processing gaze following outputs, which struggles to capture whether the gaze points for a pair of people falls on the same semantic item.

In addition, we note that better gaze following performance does not always translate to better social gaze performance. For instance, although $\text{Chong}_S$* has a better distance score compared to $\text{Chong}_T$*, it performs worse for shared attention. This effect is even more pronounced on ChildPlay (Supp C), and suggests the benefit of considering social gaze metrics for better characterizing the performance

**(a) Results on GazeFollow (41).**

| Method | Multi | AUC↑ | Avg.Dist.↓ | Min.Dist.↓ |
|---|---|---|---|---|
| Fang (14) | ✗ | 0.922 | 0.124 | 0.067 |
| Tonini (50) | ✗ | 0.927 | 0.141 | - |
| Jin (23) | ✗ | 0.920 | 0.118 | 0.063 |
| Bao (4) | ✗ | 0.928 | 0.122 | - |
| Hu (21) | ✗ | 0.923 | 0.128 | 0.069 |
| Tafasca (48) | ✗ | 0.936 | 0.125 | 0.064 |
| Chong$_S$ (9) | ✗ | 0.921 | 0.137 | 0.077 |
| Gupta (17) | ✗ | 0.933 | 0.134 | 0.071 |
| Jin (22) | ✓ | 0.919 | 0.126 | 0.076 |
| Ours-static | ✓ | 0.929 | **0.116** | **0.059** |

**(b) Results on VAT (9).**

| Method | Multi | Dist.↓ | AP$_{IO}$ ↑ |
|---|---|---|---|
| Fang (14) | ✗ | 0.108 | 0.896 |
| Tonini (50) | ✗ | 0.129 | - |
| Jin (23) | ✗ | 0.109 | 0.897 |
| Bao (4) | ✗ | 0.120 | 0.669 |
| Hu (21) | ✗ | 0.118 | 0.881 |
| Tafasca (48) | ✗ | 0.109 | 0.834 |
| Chong$_T$ (9) | ✗ | 0.134 | 0.853 |
| Gupta (17) | ✗ | 0.134 | 0.864 |
| Jin (22) | ✓ | 0.134 | **0.880** |
| Ours | ✓ | **0.105** | 0.869 |
| Ours† | ✓ | **0.105** | 0.869 |

**(c) Results on ChildPlay (48).**

| Method | Multi | Dist.↓ | AP$_{IO}$ ↑ |
|---|---|---|---|
| Tafasca (48) | ✗ | 0.107 | 0.986 |
| Gupta (17) | ✗ | 0.113 | 0.983 |
| Ours | ✓ | 0.117 | **0.994** |
| Ours† | ✓ | **0.113** | 0.993 |

**(d) Results on UCO-LAEO (31).**

| Method | Dist.↓ | AP$_{LAEO}$ ↑ |
|---|---|---|
| Jiminez (31) | - | 0.795 |
| Doosti (11) | - | 0.762 |
| Jiminez (30) | - | 0.867 |
| Ours | 0.023 | 0.963 |
| Ours† | **0.019** | **0.974** |

Table 3: Comparison against task specific methods fine-tuned on individual datasets. Best multi-person results are in bold, overall best results are underlined. Multi indicates multi-person (✓) vs single-person (✗) gaze following methods. Ours is initialized from training on GazeFollow, while Ours† is initialized from training on VSGaze.

of gaze following models, especially its semantic performance. In the supplementary C, we provide a breakdown of performance on each of the component datasets of VSGaze.

**State-of-the-art comparison: fine tuning on individual datasets.** Table 3 compares our model against task specific methods. For GazeFollow, we use our static model (Ours-static) that was trained on GazeFollow and used to initialize our model trained on VSGaze. For the video datasets, as SoTA methods were trained (or finetuned) on individual datasets, for fairness we also fine-tune our model on these datasets, investigating two initialization alternatives: either from the model trained on GazeFollow (Ours), or from the model trained on VSGaze (Ours†). Note that we are unable to compare against previous results for VideoCoatt due to our new pair-wise evaluation protocol that better captures SA performance (Sec 4.3).

On GazeFollow and VAT, our model outperforms the only other comparable multi-person gaze following model of Jin (22). It also achieves competitive or better results to single-person methods, even those leveraging auxiliary modalities such as depth (14; 50; 4; 23; 21; 48). Importantly, on the social LAEO task, we set the new state of the art on UCO-LAEO, far outperforming methods designed specifically for LAEO (31; 30; 11).

We also note that fine-tuning using the VSGaze model initialization can improve results compared to the standard protocol of fine-tuning after training on GazeFollow (ex. distance on ChildPlay and AP$_{LAEO}$ on UCO-LAEO). This suggests that training on VSGaze can leverage the complementary knowledge provided by the different tasks and datasets, which follows observations made in other works addressing multi-task training (10).

## 5.2 Analysis

**Impact of Architecture.** Comparing the performance of our model with no social gaze losses (Ours-noSoc) against the baselines (Table 2), we see that it already performs on par or better than them while being much more efficient as it processes the image only once for all people in the scene. It also serves as a strong gaze following baseline to compare performance against.

**Impact of Social Gaze Loss.** Our architecture can further benefit from the social gaze losses, showing improved gaze following performance and social gaze prediction (Ours and Ours-PP, Table 2). In particular, we observe significant gains for the SA compared to Ours-noSoc. Interestingly, the addition of the social gaze losses also better aligns the gaze following outputs for social gaze prediction. Comparing Ours-PP and Ours-noSocial, we see that performance for all social gaze tasks is improved.

**Impact of Gaze Following Loss.** We additionally train our model without the standard gaze following losses: heatmap, gaze vector and in-out (Ours-noGF, Table 2). Across VSGaze, we see that

performance for all social gaze tasks drops, which indicates that the gaze following and social gaze losses provide complementary information, and using both can give improved performance.

**Impact of VSGaze.** When comparing the performance of the models trained on VSGaze (Supp Table 4) against their versions fine-tuned on individual datasets (Table 3), we see that the fine-tuned models always perform better. This is because fine-tuning allows the models to learn dataset specific priors (ex. more LAH cases in VAT, Table 1). For instance, on VAT, Gupta* has a distance score of $0.138$ when trained on VSGaze, compared to a score of $0.134$ when directly fine-tuned on VAT. Also, our model has a distance score of $0.112$ when trained on VSGaze, and a score of $0.105$ when fine-tuned on VAT. This highlights the challenge in leveraging multiple datasets: while we may expect better performance by having more data, the different priors and statistics bring additional difficulties. Nevertheless, our model trained on VSGaze is able to achieve strong performance across all datasets.

**Impact of temporal information.** Comparing the static and temporal versions of our model trained on VSGaze, we observe improvements in performance for shared attention ($0.555$ vs $0.576$, Tab. 5 in supplementary), and similar or slightly improved performance for other metrics. This is in contrast with $\text{Chong}_T$, which often has lower performance than $\text{Chong}_S$ (for distance, LAH and LAEO metrics in Table 2). These results follow observations from prior work (Section 2) and highlight the challenge in leveraging temporal information for gaze following. We provide a detailed analysis in supplementary D.

**The supplementary** (overview in A) provides additional ablations and discussions.

## 6 Conclusion

We propose a new framework for multi-person, temporal gaze following and social gaze prediction comprising of a novel architecture and dataset. Through a series of experiments, we show that our model can effectively learn from a mix of video-based datasets with different statistics, to perform gaze following and social gaze prediction without sacrificing performance on any of them. The trained model can then be further fine-tuned on individual datasets to improve performance towards a specific scenario or task.

We hope that our proposed framework opens new directions for modeling people's gaze behavior, and are keen to see applications of the proposed dataset and social gaze losses in other methods. In the future, we intend to further investigate the benefits of temporal information and other auxiliary signals, including new ways of incorporating them into the architecture. We also plan to expand the VSGaze dataset to include more samples and annotations.

**Acknowledgement.** This research has been supported by the AI4Autism project (Digital Phenotyping of Autism Spectrum Disorders in Children, grant agreement number CRSII5 202235/1) of the the Sinergia interdisciplinary program of the SNSF.

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

## A   Supplementary Material

The supplementary material is organized as follows:

- B: details on the construction of VSGaze;
- C: breakdown of performance on VSGaze by component dataset;
- D: ablations on temporal window length and novel architecture components;
- E: implementation details;
- F: experiments on incorporating auxiliary information (speaking status);
- G: qualitative analysis of predictions;
- H: details on our DPT based gaze heatmap decoder;
- I: discussion on limitations;
- J: discussion on broader impact.

Unless specified, all experiments and visualizations in the supplementary use LAH and LAEO predictions obtained via the post-processing strategy, and SA predictions from the corresponding decoder. This is because post-processing gaze following outputs for LAH and LAEO ensures that outputs for these three tasks align. We also observe slightly better performance for LAH and LAEO using the post-processing approach compared to using the predictions from their respective decoders (Table 2, Ours-PP vs Ours).

## B   VSGaze Construction Details

We provide the detailed protocol for obtaining gaze following and social gaze annotations on VSGaze:

**Gaze Target Point.** For people sharing attention in VideoCoAtt (12), we compute their gaze points as the center of the SA object's bounding box. Similarly, for a person pair LAEO in UCO-LAEO (31), we compute their gaze points as the center of the other person's head bounding box.

**LAH.** We generate LAH annotations for all datasets. To do so, similarly to Tafasca *et al*. (48), we check whether the gaze point for an annotated person falls inside any other person's head bounding box. For the GazeFollow test set, at least 2 of the annotated gaze points should fall inside another person's head bounding box.

**LAEO.** We use the provided annotations for UCO-LAEO. For VAT and ChildPlay, we generate LAEO annotations by using the LAH annotations, checking whether the LAH target for a pair of people corresponds to the other person. We cannot obtain LAEO for GazeFollow as most images are annotated for a single person, and for VideoCoAtt because if a person is an SA target, they are not in the set of people sharing attention and their gaze is not annotated.

**SA.** We use the provided annotations for VideoCoAtt. For VAT and ChildPlay, we generate novel SA annotations from the LAH annotations, checking whether two person share their attention to the same third person. We cannot obtain SA for GazeFollow as most images are annotated for a single person, and for UCO-LAEO as a pair of people annotated with LAEO cannot be sharing attention.

## C   Breakdown of Results on VSGaze

We provide a breakdown of results on VSGaze by component dataset in Table 4. Note that following the results in the main paper, LAH and LAEO results for Ours-noGF and Ours are obtained from their respective decoders. We can observe that performance trends on individual datasets can differ from the aggregated results on VSGaze. For instance, although we have a small improvement for LAH compared to the baselines across VSGaze, we perform significantly better on ChildPlay. In general, as VideoCoAtt represents the highest number of samples in VSGaze, it also has the highest impact.

Also on ChildPlay, once again we see that better gaze following performance does not translate to better social gaze performance. Although Gupta* has a better distance score compared to Chong$_S$*, it performs significantly worse for all social gaze tasks.

## D   Ablations

### D.1   Temporal Window Length

We compare performance of our model for different temporal window lengths on VSGaze in Table 5. Note that $T = 1$ corresponds to a static model. We observe that incorporating temporal information

| Dataset | Method | PP | Dist. ↓ | AP$_{IO}$ ↑ | F1$_{LAH}$ ↑ | F1$_{LAEO}$ ↑ | AP$_{SA}$ ↑ |
|---|---|---|---|---|---|---|---|
| VAT (9) | Chong$_S$* (9) | ✓ | 0.132 | 0.798 | 0.785 | 0.486 | 0.288 |
| | Chong$_T$* (9) | ✓ | 0.137 | 0.843 | 0.783 | 0.479 | 0.332 |
| | Gupta* (17) | ✓ | 0.138 | 0.795 | 0.766 | 0.518 | 0.300 |
| | Ours-noGF | ✗ | - | - | 0.766 | 0.503 | 0.435 |
| | Ours-noSoc | ✓ | 0.121 | **0.847** | 0.812 | **0.557** | 0.440 |
| | Ours | ✗ | **0.112** | 0.845 | 0.791 | 0.526 | **0.521** |
| | Ours-PP | ✓ | **0.112** | 0.845 | **0.825** | 0.548 | 0.497 |
| ChildPlay (48) | Chong$_S$* (9) | ✓ | 0.123 | 0.973 | 0.597 | 0.470 | 0.154 |
| | Chong$_T$* (9) | ✓ | 0.137 | 0.985 | 0.572 | 0.416 | 0.165 |
| | Gupta* (17) | ✓ | 0.119 | 0.979 | 0.571 | 0.428 | 0.132 |
| | Ours-noGF | ✗ | - | - | 0.609 | 0.404 | 0.207 |
| | Ours-noSoc | ✓ | 0.118 | **0.994** | 0.620 | 0.412 | 0.188 |
| | Ours | ✗ | **0.113** | 0.993 | **0.682** | 0.426 | 0.179 |
| | Ours-PP | ✓ | **0.113** | 0.993 | 0.651 | **0.436** | 0.216 |
| VideoCoAtt (12) | Chong$_S$* (9) | ✓ | 0.120 | - | 0.793 | - | 0.290 |
| | Chong$_T$* (9) | ✓ | 0.126 | - | 0.790 | - | 0.337 |
| | Gupta* (17) | ✓ | 0.115 | - | 0.815 | - | 0.347 |
| | Ours-noGF | ✗ | - | - | 0.733 | - | 0.524 |
| | Ours-noSoc | ✓ | 0.107 | - | 0.822 | - | 0.335 |
| | Ours | ✗ | **0.106** | - | 0.804 | - | **0.601** |
| | Ours-PP | ✓ | **0.106** | - | **0.825** | - | 0.345 |
| UCO-LAEO (31) | Chong$_S$* (9) | ✓ | 0.031 | - | 0.986 | 0.811 | - |
| | Chong$_T$* (9) | ✓ | 0.064 | - | 0.941 | 0.774 | - |
| | Gupta* (17) | ✓ | 0.031 | - | 0.989 | 0.859 | |
| | Ours-noGF | ✗ | - | - | 0.989 | **0.939** | - |
| | Ours-noSoc | ✓ | 0.043 | - | 0.978 | 0.840 | - |
| | Ours | ✗ | **0.027** | - | 0.990 | 0.888 | - |
| | Ours-PP | ✓ | **0.027** | - | **0.994** | 0.870 | - |
| **VSGaze** | Chong$_S$* (9) | ✓ | 0.121 | 0.918 | 0.778 | 0.562 | 0.288 |
| | Chong$_T$* (9) | ✓ | 0.130 | **0.956** | 0.764 | 0.529 | 0.331 |
| | Gupta* (17) | ✓ | 0.119 | 0.929 | 0.784 | 0.590 | 0.335 |
| | Ours-noGF | ✗ | - | - | 0.738 | 0.579 | 0.515 |
| | Ours-noSoc | ✓ | 0.111 | 0.945 | 0.802 | 0.598 | 0.339 |
| | Ours | ✗ | **0.107** | 0.940 | 0.795 | 0.590 | **0.576** |
| | Ours-PP | ✓ | **0.107** | 0.940 | **0.812** | **0.603** | 0.352 |

Table 4: Comparison against gaze following methods on VSGaze and its component datasets: VAT (9), ChildPlay (48), VideoCoAtt (12) and UCO-LAEO (31). All models were trained on VSGaze. PP indicates social gaze predictions from post-processing gaze following outputs (✓) vs predictions from decoders (✗). Best results are in bold, second best results are underlined.

can improve performance, especially in the case of shared attention. For the other metrics performance remains comparable. As a temporal window of 9 does not necessarily give better performance than a temporal window of 5, we use $T = 5$ for our experiments.

We note that these observations are in contrast to those from the static (Chong$_S$) and temporal models (Chong$_T$) of (9). As seen in Table 2, Chong$_T$ often performs worse than Chong$_S$, with especially lower scores for the distance and LAEO metrics. This follows prior observations from the state of the art regarding temporal modelling for gaze following (Section 2), and illustrates the challenge in leveraging temporal information.

While architecture design may be a reason for the lack of greater improvement in performance, the data itself is an important factor. Firstly, despite the larger number of samples in VSGaze compared to standard video based gaze datasets, there is high redundancy between frames so data diversity is not comparable to that of GazeFollow. Secondly, the moments where temporal information is important, such as during gaze shifts, only form a small percentage of total instances. Hence, improvements for predictions in these moments are not reflected in overall metrics. For instance, gaze shifts form less than 10% of total instances in ChildPlay (48). However, in our qualitative analysis (Section G) we can see situations where temporal information helps. In future work we plan to investigate new metrics for evaluating the performance of temporal models.

### D.2 Architecture Components

We systematically remove different novel components of our architecture to analyse their impact and provide results in Table 6.

**Interaction Module.** Removing the Person-to-Scene Interaction encoder $\mathcal{I}_{ps}^{b}$ (Ours-noI$_{ps}$) has the

| $T$ | Dist. $\downarrow$ | $\text{AP}_{\text{IO}}$ $\uparrow$ | $\text{F1}_{\text{LAH}}$ $\uparrow$ | $\text{F1}_{\text{LAEO}}$ $\uparrow$ | $\text{AP}_{\text{SA}}$ $\uparrow$ |
|---|---|---|---|---|---|
| 1 | 0.108 | **0.946** | 0.806 | 0.599 | 0.555 |
| 5 | 0.107 | 0.940 | **0.812** | **0.603** | **0.576** |
| 9 | **0.106** | 0.943 | 0.811 | 0.590 | 0.563 |

Table 5: Ablations for different temporal window lengths $T$ on VSGaze. Best results are in bold, second best results are underlined.

| Method | Dist. $\downarrow$ | $\text{AP}_{\text{IO}}$ $\uparrow$ | $\text{F1}_{\text{LAH}}$ $\uparrow$ | $\text{F1}_{\text{LAEO}}$ $\uparrow$ | $\text{AP}_{\text{SA}}$ $\uparrow$ |
|---|---|---|---|---|---|
| Ours-noI$_{ppt}$ | 0.108 | 0.937 | 0.806 | **0.612** | 0.547 |
| Ours-noI$_{sp}$ | **0.105** | 0.936 | **0.813** | 0.581 | 0.545 |
| Ours-noI$_{ps}$ | 0.112 | 0.939 | 0.799 | 0.605 | 0.538 |
| Ours-noDPT | 0.111 | **0.941** | 0.810 | 0.587 | 0.528 |
| Ours | 0.107 | 0.940 | 0.812 | 0.603 | **0.576** |

Table 6: Ablations on different novel components of our architecture on VSGaze. I$_{ppt}$ refers to the Spatio-Temporal Social Interaction component, I$_{sp}$ refers to the Scene-to-Person encoder, I$_{ps}$ refers to the Person-to-Scene encoder and DPT refers to the gaze heatmap decoder. Best results are in bold, second best results are underlined.

largest impact on performance, especially for distance, LAH and SA. Without this encoder, the frame tokens cannot access the person tokens, so encoding their gaze relevant salient items is much harder. Removing the Scene-to-Person Interaction encoder $\mathcal{I}_{\text{sp}}^{\text{b}}$ (Ours-noI$_{sp}$) decreases LAEO and SA performance. Without this encoder, the person tokens cannot access the frame tokens, so they cannot capture the locations of gazed at salient items. As the frame tokens may be able to adapt to this change, gaze following performance is not impacted negatively. Finally, removing the Spatio-Temporal Social Interaction component $\mathcal{I}_{\text{pp}}^{\text{b}}$, $\mathcal{I}_{\text{pt}}^{\text{b}}$ (Ours-noI$_{ppt}$) decreases LAH and SA performance. Without this component, there is no interaction between person tokens, so identification of social dynamics is hindered. Interestingly, we see a boost in LAEO performance. However, a major portion of LAEO positives come from the UCO-LAEO dataset (see Table 1) which consists mainly of two person scenes, so capturing social interactions may be less important.

**DPT Decoder.** We replace our proposed modified DPT decoder for gaze heatmap prediction (Section H) with a simpler decoder (Ours-noDPT). This decoder projects the frame and person tokens from the last Interaction block, performs a dot product between them, and then upscales the output to the heatmap resolution. Using the simpler decoder results in drops in performance for the distance, LAEO and SA metrics. Unlike the DPT, it lacks multi-scale representations which impacts heatmap prediction and supervision of tokens.

Overall, we observe that removing the components tends to impact shared attention performance. This is similar to the observation in the previous section regarding temporal information. Unlike LAH and LAEO where the target is always another person, SA is more challenging as the shared attention target can be any person or object/point. Hence, this task may be benefitting more from additional information or architecture components.

# E   Implementation Details

The Gaze Backbone $\mathcal{G}_{\text{stat}}$ is a ResNet18 pre-trained on Gaze360 (25), and processes the head crops at a resolution of $224 \times 224$. We use a ViT-base model (1) $\mathcal{V}$ initialized with MultiMAE weights (3) to process the scene at $224 \times 224$. The temporal position embedding $\mathbf{x}$ is zero-initialized, while all other layers are randomly initialized. For training, we use the AdamW optimizer (28) with warmup and cosine annealing. The loss coefficients are set as $\lambda_{\text{HM}} = 1000$, $\lambda_{\text{VEC}} = 3$, $\lambda_{\text{IO}} = 2$ and $\lambda_{\text{LAH}} = \lambda_{\text{SA}} = 1$. To counter the class imbalance in positive vs negative social gaze labels, we weight positive samples for the social gaze loss by 2.

All models were trained on an internal cluster using a single Nvidia RTX 3090 GPU with 24GB memory. Training time for a single experiment is approximately 8 hours on both GazeFollow and VSGaze. Total compute across all experiments is approximately 140 GPU hours.

# F   Incorporating Auxiliary Information

Previous studies on analyzing conversations during meetings have shown that people usually look at the other speaking participants (45), and such cues can be exploited for gaze target selection

| Dataset | Method | Dist. ↓ | AP$_{IO}$ ↑ | F1$_{LAH}$ ↑ | F1$_{LAEO}$ ↑ | AP$_{SA}$ ↑ |
|---------|--------|---------|-------------|--------------|---------------|-------------|
| VSGaze  | Ours-spk | **0.107** | 0.938 | 0.810 | 0.588 | **0.590** |
|         | Ours     | **0.107** | **0.940** | **0.812** | **0.603** | 0.576 |

Table 7: Performance for incorporating people's speaking information in our model. Best results are in bold.

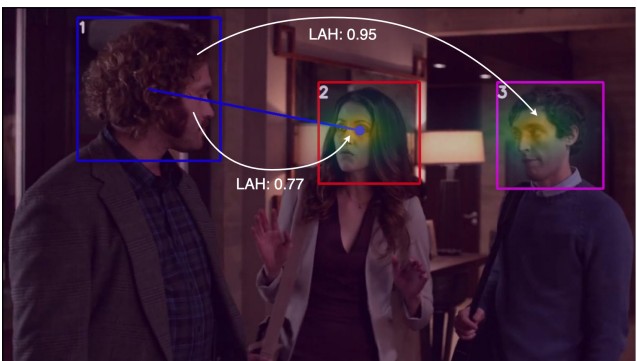

Figure 3: An illustration of the few cases where the predicted gaze point does not match with the predicted LAH label. The uncertainty in the gaze target is reflected in the heatmap, while the uncertainty in the LAH target is reflected in the LAH scores.

(35). Hence, we expect that identifying speaking persons can provide better scene understanding for gaze following, and help recognize attentiveness towards people, especially speakers. The latter is especially important in autism diagnosis, as eye contact is closely monitored by the clinician when they call out to the tested child (2).

### F.1 Experiments and Results

To incorporate speaking information in our model, we adapt the Person Module (Section 3). Specifically, we first obtain speaking scores for each person using an active speaker detection model (33). The model is retrained to detect speakers using the video alone and with no audio. The obtained speaking scores $\mathbf{s}_{i,t}$ for each person are linearly projected to the token dimension using $\mathcal{P}_{\text{spk}}$, and added to the person token. Hence, the new person token is obtained as:

$$\mathbf{p}_{i,t} = \mathcal{P}_{\text{gaze}}(\mathbf{g}_{i,t}^{\text{temp}}) + \mathcal{P}_{\text{spk}}(\mathbf{s}_{i,t}) + \mathcal{P}_{\text{box}}(\mathbf{h}_{i,t}^{\text{box}}). \tag{5}$$

We note that this formulation can easily be extended to incorporate other kinds of person-specific auxiliary information such as gestures. We explore this aspect in a follow up work where we extract gestures and other cues using vision-language models (18).

We provide results for incorporating speaking information in Table 7. On VSGaze, once again we see improvements for SA while other metrics remain similar. As the speaker detection model tends to fail in cases with side-view faces or for children in the case of ChildPlay, we may further benefit from a better speaker detection model. In addition, investigating other ways to better capture and incorporate such auxiliary information in our model is a direction for future research.

## G  Qualitative Analysis

### G.1  Qualitative Results and Comparisons

We provide qualitative results from our models in Figure 5. We observe that all models perform well, accurately capturing people's gaze targets and social gaze behaviour. We also see that incorporating temporal and speaking information can help improve predictions.

In the first sequence, the static model occasionally makes an error for person 1, and picks person 2's hands as the predicted gaze target. This is because it cannot distinguish blinking from when a person lowers their gaze. On the other hand, the temporal models recognize blinking and maintain the target as person 2's face. In the second sequence, the static model misses shared attention between persons 1,2,3 in the first frame and persons 2,3 in the third frame. This sequence is challenging due to the presence of subtle head motions which the temporal models can better capture. In the third sequence,

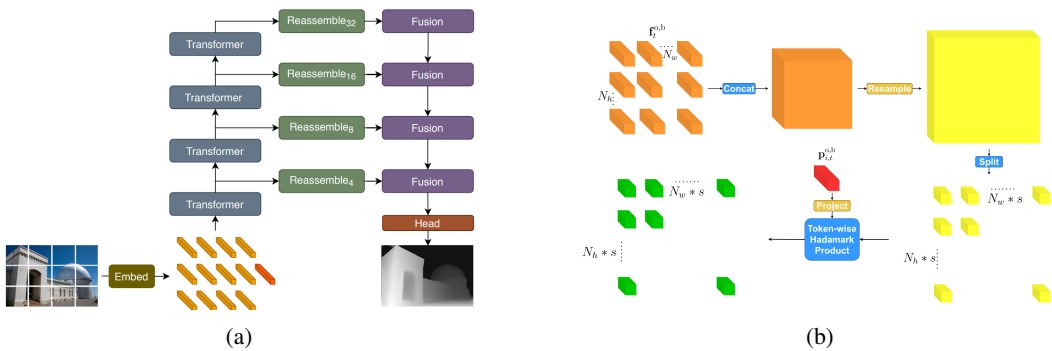

(a)              (b)

Figure 4: The standard DPT (a, taken from (39)) and our proposed person-conditioned re-assemble stage (b). This transformed DPT is used for predicting gaze heatmaps for each person in the scene.

both the static model and our proposed model make an error when predicting person 3's gaze target in frame 2. However, our model with speaking information recognizes person 1 as the target given their high speaking score.

In addition, we provide qualitative comparisons of our model against other methods in Figure 6. We see that our model performs better overall, accurately inferring people's gaze target and social gaze behaviour despite the complexity of the scenes with multiple salient targets, obscured eyes, varied settings and age groups.

### G.2 Alignment Between Gaze Following Outputs and Social Gaze Decoders

We perform analysis to understand the difference in performance between post-processing gaze following outputs or using the predictions from the task specific decoders. For LAH, we find that predictions for the two schemes align 87% of the time. On visualizing outputs, we identify that cases where they don't match are usually where the model is confused between two potential targets. So the arg max of the gaze heatmap for obtaining the gaze point picks one target, while the arg max of the LAH scores picks the other target. This confusion in target selection is illustrated for person 1 in Figure 3. We see that the predicted gaze heatmap highlights both person 2 and 3, and similarly, the LAH scores for looking at both persons 2 and 3 are high.

## H   Gaze Heatmap Prediction Details

As mentioned in Section 3, we rely on the standard DPT (39) decoder that has been developed for dense prediction, and propose an interesting way to transform it for performing person-conditioned gaze heatmap prediction.

Similar to an FPN (27), the DPT assembles the set of ViT image tokens into image-like feature representations at various resolutions. The feature representations are then progressively fused into the final dense prediction. Specifically, the DPT decoder contains two stages: 1) A *Re-assemble* stage to construct feature maps at specific resolutions at each block, and 2) a *Fusion* stage where the feature maps across consecutive blocks are upscaled and combined.

To include the person specific information, as represented by the person token in our architecture, we modify the re-assemble stage to filter only the information relevant for the given person (Figure 4). More precisely, following the standard DPT, we first project the input frame tokens $\mathbf{f}_t^{\text{o,b}}$ at level $b$ to a lower token dimension using a $1 \times 1$ conv layer, followed by spatial upsampling or downsampling using a strided $3 \times 3$ transposed conv layer or conv layer respectively;

$$\mathbf{f}_t^{\text{DPT,b}} = \text{Split}(\text{Resample}^b(\text{Concat}(\mathbf{f}_t^{\text{o,b}}))) \tag{6}$$

where Split is the reverse of the spatial concatenation operation, Concat, and Resample is the spatial upsampling/downsampling operation. To condition on a person, we then perform a token-wise hadamard product of the projected frame tokens with the the projected person token (using projection layer $\mathcal{P}_{\text{DPT}}^b$) from the corresponding block $b$.

$$\mathbf{f}_t^{\text{DPT-c,b}} = \mathbf{f}_t^{\text{DPT,b}} * \mathcal{P}_{\text{DPT}}^b(\mathbf{p}_{i,t}^{\text{o,b}}) \tag{7}$$

where $*$ denotes the hadamard product. The standard Fusion stage then follows to obtain the predicted gaze heatmap. Note that superscripts DPT, DPT-c denote updates to the frame tokens.

# I  Limitations

As discussed in Supp. G, gaze following outputs and predictions from the social gaze decoders do not always align ($13\%$ cases). One possible solution is to post-process gaze following outputs for LAH and LAEO as done in the case of Ours-PP. This ensures that outputs for gaze following, LAH and LAEO align. However, further aligning the outputs with SA is more challenging as post-processing for SA does not account for whether the gaze points fall on the same semantic item. Ensuring that outputs for all social gaze tasks and gaze following align is a challenging and interesting direction for future research. Potential directions include more refined post-processing techniques, or the addition of task consistency losses and regularization. The latter is particularly interesting as it may further increase the benefit of the social gaze losses.

Secondly, as discussed in supplementary D, temporal information does not seem to bring large improvements in performance, with improvements observed mainly for SA. Investigating new architectures, datasets and metrics to improve training and evaluation of temporal information is another important and interesting direction for future research.

# J  Broader Impact

Gaze following and social gaze prediction has strong potential for positive societal impact. Indeed, eye-contact and shared attention are monitored during the ADOS autism diagnostic test for children (2) as a marker for social communication. Given the time consuming nature of clinical tests and large scale prevalence of autism (1 in 36 children (29)), automatic tools for autism screening based on gaze can help flag potential autism cases and reduce the burden on doctors (47).

Nevertheless, as with any medical application, such screening tools must be used carefully to avoid missing cases, for instance through human-in-the-loop systems. It is also important to ensure that gaze algorithms do not invade the privacy of people. They should be deployed with appropriate consent and only for specific applications.

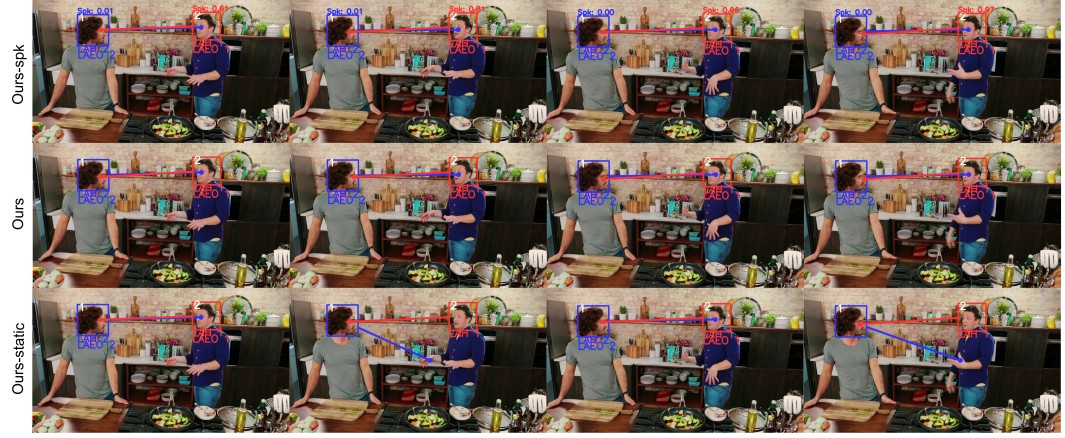

(a) Ours-static fails to recognise person 1 blinking in frames 2,4

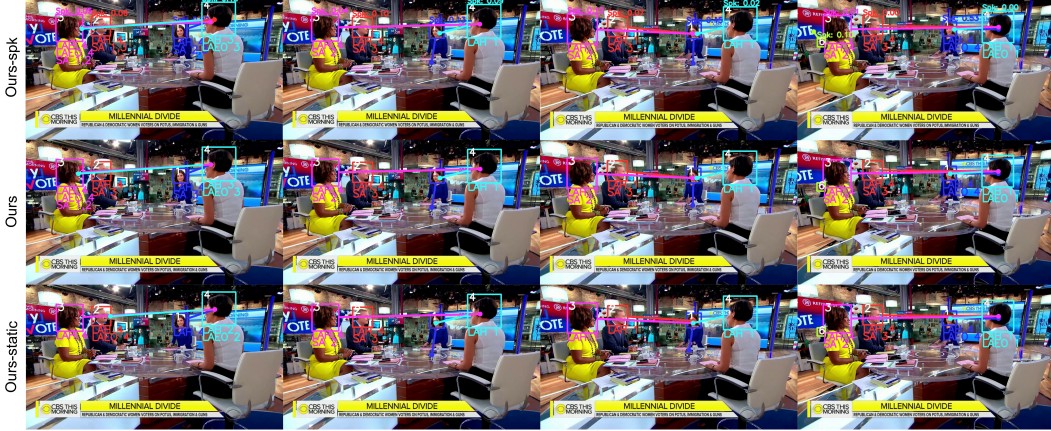

(b) Ours-static misses shared attention behaviour in frames 1,3

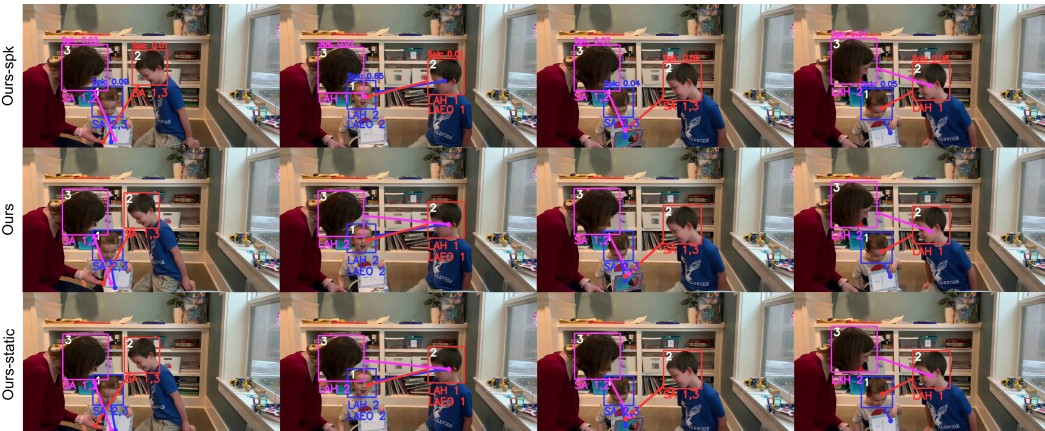

(c) Ours-spk captures the right target for person 3 in frame 2

Figure 5: Qualitative results of our proposed model (Ours), our model with speaking information (Ours-spk) and our model without temporal information (Ours-static). When the target is predicted to be inside the frame, we display the predicted gaze point and the social gaze tasks with the associated person id(s).

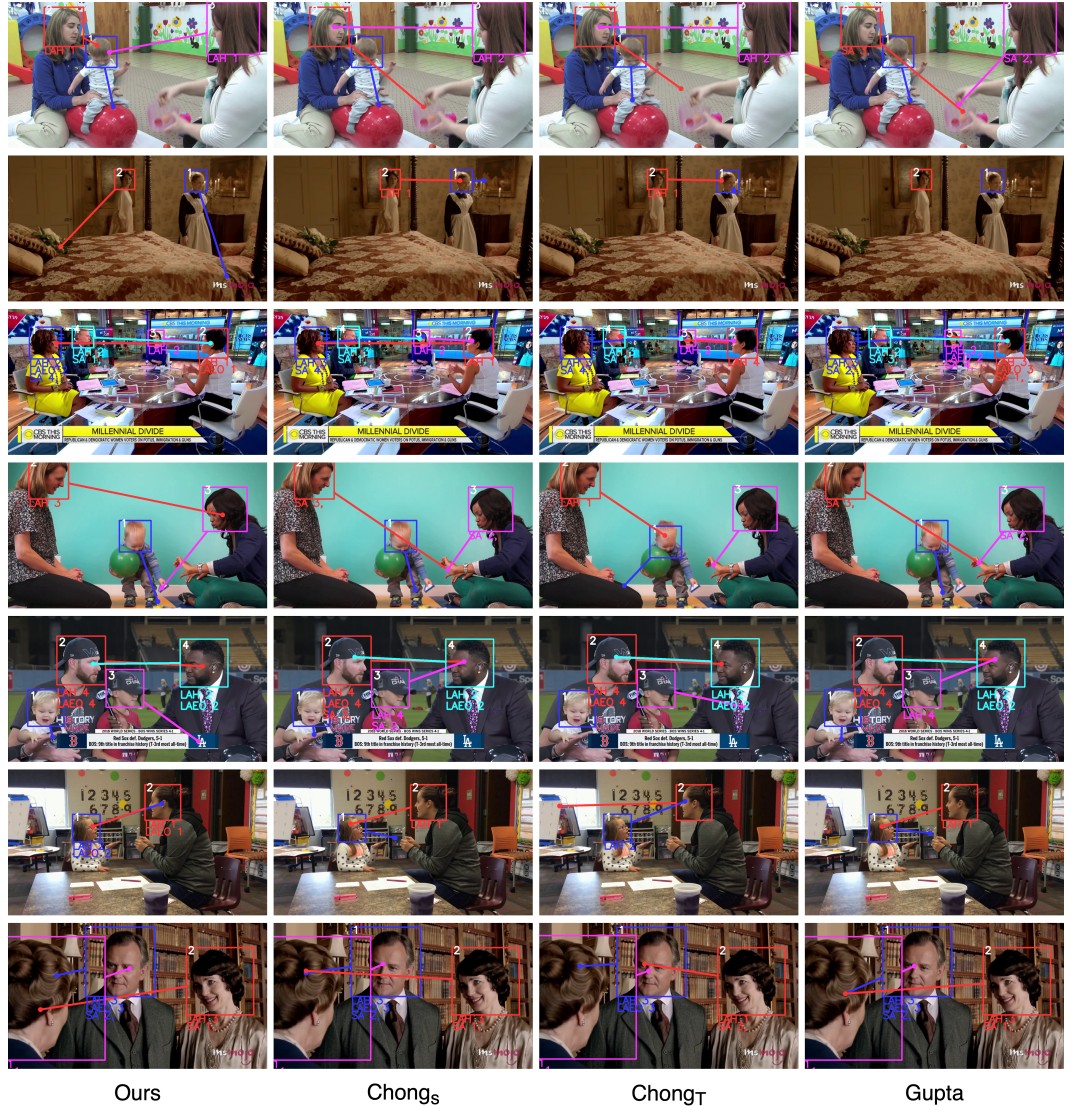

| Ours | Chong_S | Chong_T | Gupta |

Figure 6: Qualitative comparison of our model against other methods Chong$_S$, Chong$_T$ (9), Gupta (17). Our model performs better overall, outperforming other methods in complex scenes with obscured eyes, multiple salient targets, varied settings and age groups.

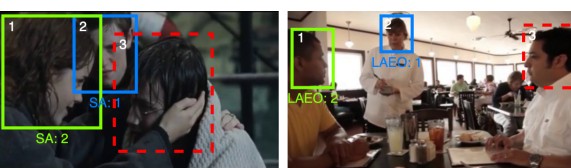
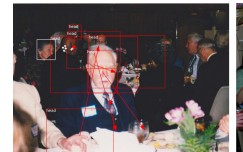

Figure 7: Samples from VACATION. Person 3 in both cases is missed as the associated person is already annotated with a social gaze 'state'.

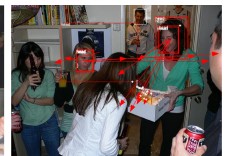

Figure 8: Results from Tonini et al. (2023) demonstrating incorrect head detections. Annotations are in white.

