# OpenReview forum: "MTGS: A Novel Framework for Multi-Person Temporal Gaze Following and Social Gaze Prediction"
_NeurIPS.cc/2024/Conference — NeurIPS 2024 poster_

### Official Review · Reviewer_FWeu · 2024-07-12

**Soundness:** 3
**Presentation:** 1
**Contribution:** 3
**Rating:** 6
**Confidence:** 5

**Summary:**

This paper handles the problem of predicting human social labels and gaze heatmap simultaneously for all people in the input image sequence, which improve the accuracy of gaze following based on human social cues. Specifically, they first calculate person tokens through a person module, and then design an interaction module to extract interactive relationship between humans and scene. The author proves the effectiveness of the proposed model on multiple datasets, and verifies the role of each part of the model through ablation experiments. And a new dataset suitable for both subtasks is also annotated.

**Strengths:**

1. The experimental section of this article is very detailed, and good results have been achieved in comparison with different methods on multiple datasets, proving the universality of this method.

2. The author for the first time incorporates human social cues into gaze prediction tasks to obtain interaction relationships between different individuals, which is an interesting topic.

**Weaknesses:**

1. The author lacks an introduction to the Pairwise Instance Generator module in the model. What is its function? There is also a lack of explanation of the number of Interaction Blocks. What impact does the change of B have on the effect of the model?
2. The symbols used in the article are confusing and difficult to understand. For example, “Each block then processes the set of output person tokens $P_{1:N_n,1:T}^{o,b-1}$  and output frame tokens $f_{1:T}^{o,b-1}$  from the previous block”, why not indicate the range of b in this sentence? And too many superscripts and subscripts in formulas and expressions can easily confuse readers in the introduction to Interaction Module.
3. Lack of visual results to prove in which scenarios the proposed model has an advantage over the comparison methods.

**Questions:**

1. Change the name of orange box in the Person Module of fig2 to Temporal Gaze Processor may be better to match the description in the essay.
2. In "$I_{ps}^b$ It is implemented as a single Transformer layer with cross-attention", "it" may be a redundant word.
3. Does  $f_t^{o,b}=V_b(f_t^{p,b})$ in line 198 denote the output frame tokens for the block b, rather than the block B in Person-Scene Interaction?
4. Where is "temporal transformer architecture" in Fig.1?

**Limitations:**

Yes

---

> ### Author Rebuttal · Authors · 2024-08-07
>
> We thank the reviewer for their feedback, and for their positive evaluation of our paper. We answer their questions and comments below.
>
> **Clarification of elements**
>
> The Pairwise Instance Generator was drawn in the figure to illustrate that the social gaze decoders take a pair of person tokens as input (L234-235). We will clarify this in the text. $f_t^{o,b}$ indeed refers to the output tokens for the specific block $b$. We use $B=4$ Interaction Module blocks as defined in L318-319. Our Interaction Module is inspired from ViT-Adaptor which found this value to give the best performance. We did not conduct additional ablations for different values of $B$. We included multiple notations for completeness when describing the Interaction Module, but will work on simplifying the notation so that it is easier to follow. The visualizations in Figure 1 are from our proposed multi-person temporal transformer model (Ours in results). We also thank the reviewer for pointing out the typos, we will correct them in the final version of the paper.
>
> **Qualitative comparisons**
>
> We provide qualitative comparisons of our method against other methods in the attached pdf with the overall response. We see that our model performs better overall, accurately capturing people’s gaze target and social gaze behavior. This is despite the complexity of the scenes with obscured eyes, multiple salient targets, varied settings (indoor, outdoor) and different age groups (children, adults). We will include these examples in the final version of the paper.

---

> > ### Comment · Reviewer_FWeu · 2024-08-13
> >
> > Thanks for your response. I will keep my rating.

---

> > > ### Author Response · Authors · 2024-08-13
> > >
> > > Thank you for the response! We appreciate your feedback and positive evaluation of our work.

---

### Official Review · Reviewer_poG5 · 2024-07-12

**Soundness:** 3
**Presentation:** 4
**Contribution:** 3
**Rating:** 6
**Confidence:** 4

**Summary:**

Paper proposes a novel framework which solves multiple gaze prediction tasks, gaze heatmap, in-out frame classification, social gaze classification for multi-person in one-pass simultanously. It also contributes a new dataset by extending existing datasets annotations. Comprehensive experiments were conducted for multiple datasets (GazeFollow, VideoAttentionTarget, ChildPlay, VideoCoAtt, UCO-LAEO which shows superior results compare to existing methods.

**Strengths:**

1. Paper's experimental setup is comprehensive and supports the central claim of the paper of unifying several gaze prediction/classifcation tasks.

2. Sufficient ablation studies were also conducted to investigate the contribution and importance of the various submodules.

3. The extension of existing datasets with more annotations is also a good contribution to the community.

**Weaknesses:**

Paper's technical contribution is incremental. The framework design, while novel, does not have any significant improvements over prior works.

The contributed dataset is also an extension of existing datasets. However, as pointed out in the rebuttal, the effecient construction process of the dataset is also a strength of this paper.

This is a good paper.

**Questions:**

One of the paper's claimed contribution is the temporal aspect of gaze. However, in the proposed model, there is no clear design of this beyond using a simple sequence of frame tokens and temporal information is aggregated with standard attention mechanism. The main discussion of temporal information is relegated to the Appendix. Will the author consider reframing the importance of temporal information in the title and abstract, or to move the temporal information discussion to the main paper?

---

> ### Author Rebuttal · Authors · 2024-08-07
>
> We thank the reviewer for their feedback, and for noting our submission as a good paper. We answer their questions and comments below.
>
> **Novelty of method**
>
> We refer the reviewer to our discussion on the novelty of our architecture in the overall response.
>
> **Dataset is an extension of existing datasets**
>
> We appreciate the reviewer highlighting our dataset as a good contribution to the community. We would like to clarify however, that while VSGaze directly benefits from existing datasets (e.g. by having gaze following annotations), it is not just a concatenation and straightforward extension of these datasets.
>
> Indeed, annotating videos with gaze following and social gaze labels is a time-consuming process, which is why existing datasets typically annotate for a subset of these labels and are small in size. Hence, we wanted to come up with a **scalable method to build a dataset with *all of these labels*, that was the *largest* of its kind, and *diverse* in terms of scene content**. A key insight was that people’s head bounding boxes could be used as a semantic entity to unify annotations across multiple gaze datasets. However, the considered **datasets only annotated a subset of people** in the scene, whereas **all people’s head bounding boxes are required** to obtain all possible positive and negative social gaze labels. Hence, we leveraged a strong head detector to extend the head bounding box annotations in these datasets and manually verified the detections. This extended set of head bounding boxes was then coupled with the existing gaze annotations to extend and unify them across datasets. The detailed process for the construction of VSGaze is given in L280-308 and supplementary B.
>
> Further, the method for constructing VSGaze can be extended in the future to obtain more gaze annotations. For instance, we focus on people’s heads as a semantic entity for unification, but could leverage strong segmentation methods such as SAM [1] to unify gaze annotations using other semantic entities such as people’s hands. Also, future dataset annotation efforts can focus on a subset of gaze annotations such as gaze following labels, and can then follow our method to extend them with social gaze annotations.
>
> **Temporal information is a simple aggregation of frame tokens**
>
> We would like to clarify that temporal information is **incorporated in our architecture by aggregating *person tokens* across time using self-attention**. While self-attention itself is fairly standard and has shown strong performance for temporal tasks such as action recognition [2], designing how to use it for a specific task and domain is a research question. Previous works on temporal gaze following [3,4] performed a frame level aggregation of temporal information, achieving limited to no success. On the other hand, **we hypothesized that the broader scene tends to remain relatively static for a short temporal window**, and instead focused on **modeling person-level temporal information** to account for **gaze direction as well as gaze target dynamics**. To do this, we incorporate temporal aggregation of person tokens at multiple levels of the architecture. As discussed in the paper (supplementary D.1) and the overall response, our architecture benefits from the addition of temporal information, with most improvements for shared attention in particular (Table 5). It also learns to account for behaviors such as blinking (Figure 5) which are not captured by current metrics. Nevertheless, more research, datasets and metrics are needed to fully exploit this information.
>
> **Moving discussion on temporal information to the main paper**
>
> We moved the discussion on temporal information to the supplementary due to space limitations. Based on the reviewer’s suggestion, we plan to move the discussion to the main paper by instead moving Section 4.2 (training and validation details) and parts of Section 4.1 (VSGaze construction details) to the supplementary.
>
> [1] Kirillov et al. (2023). Segment Anything. ICCV.
>
> [2] Tong et al. (2022). VideoMAE: Masked Autoencoders are Data-Efficient Learners for Self-Supervised Video Pre-Training. NeurIPS.
>
> [3] Chong et al. (2020). Detecting attended visual targets in video. CVPR.
>
> [4] Miao et al. (2023). Patch-level gaze distribution prediction for gaze following. WACV.

---

### Official Review · Reviewer_5XsF · 2024-07-12

**Soundness:** 3
**Presentation:** 3
**Contribution:** 3
**Rating:** 6
**Confidence:** 4

**Summary:**

This paper presents a novel framework for multi-person temporal gaze following and social gaze prediction. The authors propose an architecture that jointly predicts gaze targets and social gaze labels for all people in a scene. It uses a transformer-based model that processes both frame tokens and person-specific tokens to capture gaze information and interactions. They also build a dataset called VSGaze, which unifies and extends annotations across multiple existing gaze following and social gaze datasets. This allows joint training on multiple tasks and datasets. The proposed model achieves state-of-the-art results for multi-person gaze following and competitive performance on social gaze prediction tasks.

**Strengths:**

- Overall, this is a very solid study, and the comprehensive approach of organizing the data and training a large-scale unified model is quite logical.
- There is a noticeable improvement in performance across multiple tasks. Experiments are being conducted thoroughly from various angles.

**Weaknesses:**

- Conversely, the network's structure is relatively straightforward, combining existing approaches, and its novelty as a method is not necessarily significant.
- The model's generalization capability to unknown domains and datasets has not been evaluated. While preparing the data is challenging, it is particularly crucial for methods involving humans.

**Questions:**

It seems challenging to clearly state whether the training of models combining multiple tasks is useful even with a small amount of data based on the current results. Is there anything we can say about the relationship between data quantity and joint training? Since the effect of large-scale training with multiple datasets itself is not the core argument of this paper, I think it would be more insightful if this aspect can be carefully separated from the observations.

**Limitations:**

The discussion in the appendix seems to be conducted appropriately.

---

> ### Author Rebuttal · Authors · 2024-08-07
>
> We thank the reviewer for their feedback, and for highlighting our paper as a very solid study. We answer their questions and comments below.
>
> **Novelty of method**
>
> We refer the reviewer to our discussion on the novelty of our architecture in the overall response.
>
> **Generalization to new domains/datasets**
>
> We agree with the reviewer that having a model that generalizes well to different domains and datasets is important. Indeed, this was a main motivation behind creating VSGaze, which extends and combines datasets containing scenes from diverse settings such as daily activities (VideoCoAtt), talk shows (VAT), childcare (ChildPlay) etc. We show that our model trained on VSGaze performs well on each of its constituent datasets in Table 4. On the other hand, training on a specific domain results in poorer generalization to other domains. For instance, we evaluated our VAT trained model on ChildPlay and found that it generalizes poorly (Dist: 0.119, $AP_{IO}$: 0.991, $F1_{LAH}$: 0.624, $F1_{LAEO}$: 0.363, $AP_{SA}$: 0.194) compared to the model trained on VSGaze (Dist: 0.113, $AP_{IO}$: 0.993, $F1_{LAH}$: 0.651, $F1_{LAEO}$: 0.436, $AP_{SA}$: 0.216). We leave the investigation of generalization to unseen domains and datasets for future work.
>
> **Data quantity vs joint training**
>
> This is an interesting and important question that we discuss under ‘Impact of VSGaze’ in L404-410. We find that fine-tuning models on specific datasets (Table 3) typically results in better performance compared to training on VSGaze (Table 4). This is because the model can learn dataset specific priors, ex. more LAH cases in VAT compared to ChildPlay (Table 1). For instance, on VAT, Gupta [1] has a distance score of 0.138 when trained on VSGaze, compared to a score of 0.134 when directly fine-tuned on VAT. Also, our proposed model has a distance score of 0.112 when trained on VSGaze, and a score of 0.105 when fine-tuned on VAT. Hence, while we may expect models to benefit from more data, accounting for different priors and statistics across datasets (especially given VSGaze’s diversity as discussed above) brings additional challenges. We will detail this aspect more in the final version of the paper.
>
> [1] Gupta et al. (2022). A modular multimodal architecture for gaze target prediction: Application to privacy-sensitive settings. CVPRW.

---

### Official Review · Reviewer_7cdN · 2024-07-12

**Soundness:** 3
**Presentation:** 3
**Contribution:** 2
**Rating:** 6
**Confidence:** 4

**Summary:**

This paper focuses on social gaze prediction in videos. An approach based on ViT has been proposed, combining three modules including person module, interaction module, and prediction module. The authors also summarised the current shortcomings of the existing datasets and have introduced a new dataset comprising social gaze interactions such as looking at someone, looking each other, and shared attention.

**Strengths:**

The main strengths of the paper are:

- Focusing on video-level gaze prediction, particularly on events like shared attention, which goes beyond gaze target estimation in images.
- Introducing a new dataset for the target problem.
- Overall, the paper is well-written and well-organized.

**Weaknesses:**

The main weakness of the paper is the results. If you refer to Tables 2 and 3, the differences in terms of distance and other metrics are very small, making it difficult to assess the benefits of the proposed approach. In some cases, the differences in terms of distance range between 0.01 and 0.001. This makes the results questionable in my view. The authors may need to run statistical tests to demonstrate that such differences are significant, or these improvements may be by chance given that the differences are negligible in most cases.

Regarding the methodology, I am not sure if papers like [1] and [2] were available at the time of submitting this paper, but it is hard to see how the proposed approach is competitive compared to these aforementioned approaches in terms of both methodological novelty and performance.

On page 3, the authors mention ".. both methods do not address the gaze following task." However, the proposed method and dataset do not address the gaze following task either. Indeed, the new dataset only includes a subset of the gaze communication cues presented by Fan et al. at ICCV 2019. Why hasn't this dataset been considered in the evaluations given its relevance?

[1] Sharingan: A Transformer Architecture for Multi-Person Gaze Following
[2] A Unified Model for Gaze Following and Social Gaze Prediction

**Questions:**

1. Could you please demonstrate that the differences between the obtained values are significant?

2. Could you please discuss your contributions with respect to the existing transformer-based approaches for gaze following and social gaze estimation?

3. Could you please explain why you selected these methods for comparison specifically, and why you chose the datasets like GazeFollow and VideoAttention, most of which do not even have annotations for social gaze cues?

**Limitations:**

The authors have discussed the limitations and broader impact satisfactorily. In addition, they have provided detailed visualisations to offer insight into failure cases.

---

> ### Author Rebuttal · Authors · 2024-08-07
>
> We thank the reviewer for their feedback, and for raising valuable discussion elements. We answer their comments and questions below.
>
> **Significance of improvements**
>
> We agree with the reviewer that differences of 0.001 can be negligible, however, differences in 0.01 are significant as discussed below:
> - Distance: This metric for gaze following computes the distance between the ground truth and predicted gaze point on a normalized 1x1 grid. For an HD image of size 1080x720, a difference of 0.01 results in a difference of up to 11 pixels in image space on average (and often more). This can result in selecting a completely different target, ex. a nearby face.
> - F1: This metric for social gaze computes the harmonic mean of the precision and recall scores. A difference of 0.01 can correspond to a difference of 3% in either precision or recall.
>
> Following the suggestion of the reviewer, we also re-trained our model with 5 different seeds on VSGaze, and obtained a standard deviation of 0.0006 for distance, 0.0017 for $AP_{IO}$, 0.0007 for $F1_{LAH}$, 0.0048 for $F1_{LAEO}$ (LAEO has significantly less positives, Table 1) and 0.0020 for $AP_{SA}$. We further include qualitative examples in the attached pdf with the overall response, where we see that our model outperforms the baselines in various complex scenes.
>
> **Comparison to existing transformer based architectures**
>
> We discuss existing transformer based architectures in L93-100, where we mention that they follow a DETR style approach for simultaneously predicting people’s head bounding box and gaze target. While their approach is interesting, they are prone to head detection errors (examples in attached pdf with overall response). And as their evaluation relies on matching detected and annotated heads (and performance is only reported on the matched heads), it is difficult to compare performance against them. Also, these methods do not address social gaze prediction.
>
> We thank the reviewer for references [1, 2], which are interesting and contemporary to our work. These were not available at the time of our submission. Nevertheless, our work differs notably from these studies.
>
> Firstly, [1] focuses solely on the gaze following task. Their architecture is based on a ViT and treats person and frame tokens equally in the self-attention layers, which as noted in their paper, can be limiting. In contrast, our architecture allows the processing of person and frame tokens through separate transformers, facilitating interactions via cross-attention. In particular, this separation also allows for temporal processing of person tokens at multiple levels of the architecture.
>
> [2] extends [1] by leveraging a frozen gaze following model, and adding graph layers to model interactions and predict social gaze. However, this approach has several drawbacks:
> - As shown in [2], the graph layers provide limited benefit, which the authors attribute to over-smoothing, a known issue in graph neural networks. Our transformer-based architecture avoids this problem and benefits from the Interaction Module components, as demonstrated in our ablations in Section D.2.
> - Since [2] freezes the gaze following backbone during training, the gaze following and social gaze tasks cannot complement each other during training. Our analysis in L393-403 demonstrates that these tasks provide mutual benefits, and jointly training our architecture with both gaze following and social gaze losses yields the best performance.
>
> In terms of performance, our method performs comparably to [1] for gaze following on VAT (Dist. 0.105 vs 0.107) and improves over [2] for LAEO on UCO-LAEO (AP: 0.974 vs 0.946). We are unable to compare performance for other tasks against [2] as they use different performance metrics. We leave a detailed quantitative comparison for future work.
>
> **Method does not address gaze following, reasoning behind choice of datasets and compared methods**
>
> We would like to clarify that **our method *does* address the gaze following task**. We discuss the gaze heatmap decoding step in L218-230 and provide results for gaze following using the distance metric (abbreviated Dist. in the tables). Indeed, our method **achieves the state of the art multi-person gaze following performance on standard benchmarks as indicated in Tables 3a,b,c**.
>
> We compare performance against other gaze following methods whose predictions can also be post-processed for social gaze. Fan et al. (2019) and Chang et al. (2023) do not address gaze following, which is an important task to identify the target of shared attention, and improves performance for social gaze prediction when trained jointly (L393-403). Further, recent methods for gaze following [3] have been shown to outperform task-specific social gaze models (L126-127).
>
> Also, Fan et al. (2019) predict a single social gaze ‘state’ for a person, not allowing for simultaneous social gaze behaviors ex. LAEO and SA (L137-139). This issue extends to their dataset, which annotates a single social gaze state for a person at a given moment (L276-279). In the attached pdf, we show examples from the dataset where this annotation protocol fails. These issues were also reported in [2,4].
>
> Besides this serious issue, VSGaze significantly differs from Fan (2019). **Content wise, it is much more diverse**, including scenes from TV shows, daily activities and childcare settings. It is also **more complex: in terms of number of people** in the scene (3.43 vs 2.14) and **size** (5.8x larger).
>
> [1] Tafasca et al. (2024). Sharingan: A Transformer Architecture for Multi-Person Gaze Following. CVPR.
>
> [2] Gupta et al. (2024). A Unified Model for Gaze Following and Social Gaze Prediction. FG.
>
> [3] Chong et al. (2020). Detecting attended visual targets in video. CVPR.
>
> [4] Belen et al. (2023). Temporal Understanding of Gaze Communication with GazeTransformer. Gaze Meets ML Workshop at NeurIPS.

---

> > ### Comment · Reviewer_7cdN · 2024-08-11
> > **reply: Rebuttal by Authors**
> >
> > I have raised my score to 'weak accept' after considering all the feedback and responses. However, as another reviewer also noted, I am still not fully convinced about the significant methodological novelty of this architecture compared to previous ones. Nonetheless, this is a well-executed paper, and introducing a new dataset is a plus.

---

> > > ### Author Response · Authors · 2024-08-12
> > >
> > > Thank you for the response! We appreciate you raising your score and noting our submission as a well executed paper.

---

### Author Rebuttal · Authors · 2024-08-07

We thank the reviewers for their feedback on our paper, which presents the following contributions:
- A novel **temporal, multi-person architecture** that jointly models gaze following and social gaze prediction.
- A new **dataset, VSGaze**, which extends and unifies annotations across multiple gaze following and social gaze datasets. As by far the largest and most diverse dataset of its kind, it opens new avenues for gaze modeling.
- New evaluation protocols and metrics for assessing semantic gaze following and social gaze performance.

We are pleased that the reviewers found our central premise of unifying several gaze prediction tasks into a single framework logical (5XsF). They appreciated our novel architecture that, for the first time, encodes human interaction relationships for gaze tasks (FWeu). They also appreciated our extensive experiments (5XsF, poG5, FWeu), which demonstrated that
(1) Our architecture can successfully model all tasks jointly, and that **this new architecture and joint training improved performance across multiple tasks and datasets compared to other methods**; (2) the importance and contribution of the different sub-modules in the overall performance.

They further noted the value of our proposed dataset (7cdN, poG5) as a good contribution to the community and appreciated our social gaze metrics for characterizing semantic gaze performance (7cdN). Lastly, they described our paper as well-written and organized (7cdN)

A common concern among reviewers (7cdN, 5XsF, poG5) was the novelty of our architecture. We would like to emphasize that our architecture was developed to address several complex research questions:

**(1) How to account for multiple people in the scene?** As discussed in lines 45-47, this challenge is inherently more complex than single-person gaze following, as our architecture has to process the scene only once, and to capture salient items for all individuals while retaining the ability to decode each person's gaze target. Among the cited gaze-following papers, only [1] presents a comparable multi-person architecture, but as it processes each person separately, it fails to account for interactions (see discussion in lines 89-100).

In contrast, by employing specific person tokens to encode individuals and an Interaction Module, our architecture is able to capture interactions between people and the scene (encoded as frame tokens), and has achieved state-of-the-art results for multi-person gaze following (Tables 3a, 3b, 3c).

**(2) How to jointly model multiple gaze tasks?** Predicting the gaze target for each individual while simultaneously predicting social gaze between pairs of people presents a significant challenge, as gaze targets are represented as heatmaps, whereas social gaze is a binary label. Jointly modeling these diverse tasks has not been previously attempted. Our architecture overcomes this challenge thanks again to our token-based representation. It leverages person and frame tokens for gaze heatmap prediction through the conditional DPT decoder, and pairs of person tokens for social gaze prediction through the social gaze decoders. This approach achieves strong performance across all tasks without compromising on any of them (Table 2).

**(3) How to incorporate temporal information?** Incorporating temporal information is particularly challenging due to the small size of gaze datasets. Only two other gaze-following methods [2, 3] have attempted this, with limited success and without accounting for gaze direction dynamics. In contrast, our architecture addresses this challenge by incorporating temporal information at multiple levels, from gaze direction dynamics to gaze heatmap prediction. Inspired by ViT-Adaptor [4], we freeze the ViT layers during training on VSGaze, allowing frame tokens to adapt through interactions with person tokens. This approach leverages temporal information and improves performance, especially for shared attention tasks (Table 5). Our qualitative examples (Figure 5) demonstrate that our temporal model captures behaviors such as blinking, but unfortunately such interesting behaviors are not accounted for by current metrics. Despite these advancements, as discussed in supplementary section D.1, further research, datasets, and metrics are needed to fully harness the potential of temporal information.

In the supplementary F, we show that our architecture also supports incorporating person-level auxiliary information such as their speaking status. We will include this discussion and all other feedback from reviewers in the final version of the paper.

[1] Jin et al. (2021). Multi-person gaze-following with numerical coordinate regression. FG.

[2] Chong et al. (2020). Detecting attended visual targets in video. CVPR.

[3] Miao et al. (2023). Patch-level gaze distribution prediction for gaze following. WACV.

[4] Chen et al. (2022). Vision transformer adapter for dense predictions. ICLR.

---

### Author Response · Authors · 2024-08-09
**Additional Remarks**

Dear reviewers, I hope our response answered your questions and comments, please let us know if you have any additional concerns. We appreciate your time, thank you!

---

### Decision · Program_Chairs · 2024-09-25

**Decision:**

Accept (poster)

**Comment:**

This paper proposes a framework for multi-person temporal gaze following and social gaze prediction. All the reviewers recommended acceptance of this paper. The AC checked the author response and the discussions, and agreed to accept this paper.